

# Technical Note: Accurate, reliable and high resolution air quality predictions by improving the Copernicus Atmosphere Monitoring Service using machine learning techniques

Angelo Riccio[1,2] and Elena Chianese[1]

[1]Department of Science and Technology, Parthenope University of Naples, Centro Direzionale, Isola C4, 80143, Naples (Italy)
[2]UNESCO Chair on 'Environment, Resources and Sustainable Development', Department of Science and Technology, Parthenope University of Naples, Via F. Petrarca 80, 80123, Naples, Italy

**Correspondence:** Angelo Riccio (angelo.riccio@uniparthenope.it)

**Abstract.** Starting from the regional air quality forecasts produced by the Copernicus Atmosphere Monitoring Service (CAMS), we propose a novel post-processing approach to improve and downscale results on a finer scale. Our approach is based on the combination of Ensemble Model Output Statistics (EMOS) with a spatio-temporal interpolation process performed through the Stochastic Partial Differential Equation-Integrated Nested Laplace Approximation (SPDE-INLA). Our interpolation approach

includes several spatial and spatio-temporal predictors, including meteorological variables. A use-case is provided, scaling down the CAMS forecasts on the Italian peninsula. The calibration is focused on the concentrations of several air quality pollutants ($PM_{10}$, $PM_{2.5}$, $NO_2$ and $O_3$) at daily resolution from a set of 750 monitoring sites, distributed throughout the Italian country. Our results show the key role played by conditioning variables to improve the forecast capabilities of ensemble predictions, thus allowing a net improvement of the calibration with respect to ordinary EMOS strategies. From a deterministic point

of view, the predictive model performance shows a significant improvement of the performance of the raw ensemble forecast, with an almost zero bias, significantly reduced root mean square errors and correlations almost always higher than 0.9 for each pollutant; moreover, the post-processing approach is able to significantly improve the prediction of exceedances, even for very low thresholds, such as those recently recommended by the World Health Organisation. This is particularly significant if a forecasting approach is to be used to predict air quality conditions and plan adequate human health protection measures, even

for low alert thresholds. From a probabilistic point of view, the forecast quality was verified in terms of reliability and credible intervals. After post-processing, the predictive probability density functions were sharp, and much better calibrated than the raw ensemble forecast. Finally, we present some additional outcomes based on a set of gridded (4 km × 4 km) daily maps covering the whole Italian country, for the detection of areas where pollution peaks forecasts (exceedances of the regulatory thresholds) occur.

## 1   Introduction

Outdoor air pollution induced by natural sources and human activities remains a major environmental problem of concern worldwide. Studies have shown that particulate matter, ozone and nitrogen dioxide degrade ambient air quality and cause



serious health problems to human beings (Kim et al., 2015; Kampa and Castanas, 2008; Manisalidis et al., 2020). For example, recent studies have suggested that air pollution, particularly traffic-related pollution, is associated with preterm birth and infant
mortality and the development of asthma and atopy (Khreis et al., 2017; Burbank and Peden, 2018). A joint study of the World Bank and the Institute for Health Metrics and Evaluation (World Bank, 2016) has shown how air pollution also has huge implications for world economies: approximately 5.5 million lives were lost in 2013 from diseases associated with outdoor and indoor air pollution, and with a global economic cost for those deaths of approximately US$225 billion in lost labour income and over US$5 trillion of welfare losses.

Producing reliable short-term forecasts of pollutant concentrations is a key challenge to support national authorities in their tasks related to EU Air Quality Directives, such as planning and reporting the state of air quality to the citizens. Starting from 2014, the Copernicus Atmosphere Monitoring Service (CAMS), a service implemented by the European Centre for Medium-Range Weather Forecasts (ECMWF), continuously provides air quality forecasts over Europe, supporting this task. This system is based on an ensemble of several models (Marécal et al., 2015). The different individual model results are interpolated on a
common regular $0.1° \times 0.1°$ grid over the European domain (25°W-45°E, 30°N-72°N) for the next four days at a hourly time resolution, and a median ENSEMBLE is calculated from the model outputs.

Higher spatial resolutions are achieved through smaller-scale applications, such as those used for the FORAIR-IT (Mircea et al., 2014), kAIROS (Stortini et al., 2020), PREV'AIR (Rouil et al., 2009), UK-AIR (DEFRA, 2022), or CALIOPE (Baldasano et al., 2008) systems. However, all these systems require the use of more detailed information and obviously imply the
use of much greater computational resources. On the other hand, the use of raw CAMS forecasts do not permit the reproduction of subgrid-scale features, especially close to large point emission sources. There is a reasonable expectation that even the ENSEMBLE results have limited skill under complex local-scale conditions, with an expected ensemble mean and variance correlated with the observations and the actual model uncertainty, respectively, and a persistent underestimation of the true observations and model uncertainty.

However, understanding how well pollutant concentrations can be predicted both in space and time is essential for a proper assessment of warning and alarm levels and for capturing concentrations gradients even at high spatial resolutions (Buizza et al., 2022; Chianese et al., 2018; Cohen et al., 2017; Lindström et al., 2014; Zhou et al., 2019). In the last years, there has been an increasing interest in spatio-temporal statistical models, combining ensemble predictions, data assimilation and machine learning, and these models rapidly gained attention in the air quality scientific community (Bai et al., 2018; Zhang
et al., 2012). The reason lies in the fact that hybrid models are easier to implement and do not require high computational resources, while deterministic models are often more computationally expensive and difficult to manage in terms of quality and number of input data requests (Bertrand et al., 2022; Camastra et al., 2022; Chianese et al., 2019; Taheri Shahraiyni and Sodoudi, 2016).

In this study, starting from CAMS air quality forecasts, we studied the possibility to improve the 24-hour ahead evolution of
$PM_{10}$, $PM_{2.5}$ (daily averages), $O_3$ (highest 8-hour daily maximum) and $NO_2$ (1-hour daily maximum) in Italy. This country is characterised by complex conditions for air pollution modelling due to topographic features, different geo-climatic zones and the complex mix of anthropogenic and natural sources of air pollution. Thus, post-processing of CAMS raw ensemble results





may be particularly suitable for such areas, where the results of the different models could benefit from the use of additional information for a more accurate and higher resolution estimation.

In this work, a machine learning framework was used to improve the estimation of the air quality forecast in Italy, combining the deterministic forecasts with additional spatio-temporal predictors within a statistical framework. More precisely, we designed an ensemble model output statistical framework to obtain a bias-corrected and well calibrated ensemble prediction from the CAMS suite, and then fit this calibrated ensemble prediction within a spatio-temporal hierarchical model using the Integrated Nested Laplace Approximation-Stochastic Partial Differential Equation (INLA-SPDE) approach (Rue et al., 2009).

The INLA-SPDE method is a deterministic approach for doing Bayesian inference, as opposed to Markov Chain Monte Carlo (MCMC) method which is a simulation-based approach (Gilks et al., 1995; Riccio et al., 2006), but which has been shown to provide a viable method for speeding up calculations, even for large-scale problems, without sacrificing accuracy.

     The remaining part of this paper is organised as follows. In section 2 we first introduce the input dataset and the statistical model we have chosen to analyse the pollutant concentrations, and in Section 3 the methods used to develop the post-

processing approach. Next, Section 4 discusses results, model validation and two possible applications of the model estimates for predicting threshold levels in Italy. Conclusions are reported in Section 5. More details about the implementation of the post-processing approach are given in A and B.

## 2    Data

### 2.1    The CAMS suite

CAMS provides daily analyses and forecasts of worldwide long-range transport of atmospheric pollutants as well as the air quality forecasts for the European domain updated on a daily basis. Over the global scale, CAMS provides the five-day forecasts of aerosols, atmospheric pollutants, greenhouse gases, as well as stratospheric ozone and UV-index. On the European scale, predictions are issued with a $0.1° \times 0.1°$ resolution over Europe and 10 vertical levels from the Earth surface up to 5000 m, combining data with satellite and non-satellite observations.

The CAMS ensemble prediction system started with a suite composed by seven air quality models: CHIMERE, EMEP, EURAD-IM, LOTOS-EUROS, MATCH, MOCAGE and SILAM. Starting from the end of 2019, the DEHM (Aarhus University, Denmark) and GEM-AQ (IEP-NRI, Poland) models were added. From June 2022, two additional models (MINNI, operated by ENEA, Italy, and the Barcelona Supercomputing Centre's MONARCH model), deliver their results, as well, expanding the ensemble size to eleven members. The 00:00 UTC ECMWF-IFS (Integrated Forecast System) provides the meteorological

data for the prediction of transport phenomena, and the CAMS emission database provides the input data for the simulation of emission phenomena. CAMS forecasts are available for download from the CAMS Atmosphere Data Store. The full range of forecast is guaranteed available by 08:00 UTC every day for the next four days. Marécal et al. (2015) provide the full detail on the implementation of this multi-model forecast system.



## 2.2 Training data and predictors

Our ultimate goal is to improve the CAMS forecast on the Italian peninsula. This geographic area is characterised by complex orographic and climatic conditions, including the mountain systems of the Alpine arc (to the north) and Apennines (along the entire longitudinal ridge from north to south), an extensive flat area (the Po valley) and two major islands (Sicily and Sardinia). Furthermore, the transport of desert dust in the Mediterranean region often affects PM concentration, with a significant impact on population health (Alahmad et al., 2023; Sajani et al., 2011). This variety of orographic and climatic conditions leads to

a high spatial variability of air quality conditions, which make the Italian peninsula a significant test bed for the predictive capabilities of the CAMS ensemble.

The following air quality pollutants have been considered in the present study: $PM_{10}$ and $PM_{2.5}$ (daily averages), $O_3$ (highest 8-hour daily maximum) and $NO_2$ (1-hour daily maximum). Table 1 reports the number of ground stations for each of the measured pollutants together with the area type and data coverage (defined as the percentage of monitoring stations with at

least 90% of valid data for the year 2022). These data are available from the Up-To-Date (UTD) channel of the Air Quality E-reporting system (https://www.eea.europa.eu/data-and-maps/data/aqereporting-9) of the European Environment Agency (EEA), from which they can be freely downloaded.

According to information communicated to the EEA, the Italian air quality network is made up of a total of 750 monitoring stations, unevenly distributed by area type: most of the monitoring stations are clustered around urban areas, while remote/ru-

ral areas are less represented. These monitoring stations are also unevenly distributed with respect to altitude, with most of monitoring sites below 250 m. This is not surprising at all, being most of the stations located where high concentrations are expected, i.e. at low-altitude urban or suburban sites.

**Table 1.** Details of observation stations grouped by pollutant and area type. Data coverage refers to the percentage of monitoring stations with at least 90% of valid data for the year 2022.

| Pollutant | Area type | | | Data coverage |
|---|---|---|---|---|
| | rural | suburban | urban | |
| $PM_{10}$ | 35 | 59 | 152 | 73% |
| $PM_{2.5}$ | 10 | 29 | 54 | 73% |
| $NO_2$ | 48 | 64 | 189 | 80% |
| $O_3$ | 47 | 39 | 72 | 81% |

As complementary information to the concentration of the main trace pollutants, several geographic and/or meteorological variables may have a potentially predictive role for air quality. The use of spatio-temporal predictors is by no means uncommon

in air quality modelling since they are usually exploited to capture the high frequency variability at finer spatial scales (Bertrand et al., 2022; Shtein et al., 2019; Stafoggia et al., 2020). The predictors used in this study can be classified into two different categories: 1) purely spatial predictors, and 2) spatio-temporal predictors.



The first category includes all the geographical variables which do not have a variable temporal component. For each monitoring station, we first built a circular buffer with a radius of 5000 m, comparable to the resolution of the raw CAMS predictions, and sampled the density of each purely spatial predictors within this buffer. The purely spatial predictors included in this study are: *i) resident population*, resident population in Italy extracted from the 2011 national census for each of the 366,863 demographic areas surveyed by the Italian Institute of Statistics (ISTAT); *ii) imperviousness density*, soil sealing at a pixel level and remapped as percentage of soil sealing within the buffer distance; *iii) imperviousness built-up*, percentage of building and no-building class within the sealing outline derived from the imperviousness density for 2018. The last two dataset are available from the ISPRA download centre (https://www.isprambiente.gov.it/it/attivita/suolo-e-territorio/suolo/copertura-del-suolo/high-resolution-layer), in raster form at a spatial resolution of 10 m for 2018. *iv) land cover*, Corine Land Cover (CLC) map available from the European Environment Agency as a shapefile for the year 2018. CLC data were reclassified as percentage covered by four classes (high urban development, low urban development/industrial/other artificial areas, agricultural areas, forest and semi-natural areas) within the buffer distance; *v) road density*, length of road segments by the Open Street Map database (https://download.geofabrik.de/). The road density was re-sampled in three classes: sum of the length of all roads segments, sum of the length of main roads (highways and trunks) and sum of the length of primary roads within the buffer distance.

For the spatio-temporal predictors we took into consideration several meteorological data, all retrieved by the ECMWF operational system and bi-linearly interpolated at each monitoring station location: *vi) total daily precipitation*; *vii) relative humidity*; *viii) wind speed* and *ix) wind direction*, all at 12 UTC; *x) planetary boundary layer height* at 00 UTC, and *xi) planetary boundary layer height* at 12 UTC. These data are expected to show potential predictive capabilities for air quality. For example, temperature, humidity, wind speed and direction lead to changes in pollutant concentrations (Liu et al., 2020; Zhang et al., 2015), with higher temperature and wind speed and lower relative humidity being favourable for ozone, particulate matter and nitrogen dioxide. Boundary layer height is also an important factor in air pollution formation, due to the enhanced convective activity and the scavenging of peroxy radicals (Chen et al., 2019a, b; Levi et al., 2020).

## 3 Methods

### 3.1 The post-processing approach

Ensemble systems are often associated with statistical post-processing steps to inexpensively improve their raw prediction properties (Vannitsem et al., 2021). Starting from raw CAMS data, we propose a two-stage post-processing approach, able to remove biases from the output distribution and improve the prediction properties.

The first stage is an ensemble model output statistical method (EMOS, Gneiting et al. (2005)), exploited to obtain a bias-corrected and well-calibrated ensemble. In the second stage, we embed this well-calibrated forecast into a hierarchical spatio-temporal framework, exploiting the previously listed spatial and temporal predictors.



All statistical analyses have been performed through the combined use of the *R* statistical software, version 4.2.2 (Venables et al., 2022), the Climate Data Operator (CDO), version 2.1.1 (Schulzweida, 2022), and Matlab®, version R2022b Update 3 (MATLAB, 2022), software. Some details about these two stages are given in the next two subsections.

### 3.1.1 Stage 1: The calibration of the ensemble

As discussed in Gneiting et al. (2005), the calibration stage has, as final goal, the maximisation of accuracy subject to reliability. Reliability measures the capability of the ensemble to predict unbiased estimates of the observed frequencies. In short, a reliable forecast is one for which there is correspondence between the forecast probability and the probability of occurrence. Reliability can be measured through the Talagrand histogram (Talagrand and Vautard, 1999; Hamill, 2001) or, equivalently, with the probability integral transform (PIT) histogram (Dawid, 1984; Gneiting et al., 2007). Talagrand and Vautard (1999) fully discuss the properties of the Talagrand and PIT histograms, i.e. how their shape can be used to assess when ensemble results are under/over-dispersed.

Reliability is a necessary but not sufficient condition for a valuable ensemble forecast. Another desirable condition is accuracy. An accurate forecast closely resembles the true state of the system; in particular, an ensemble is the more valuable the greater the accuracy compared to the one obtained with a naive method, such as climatology or persistence.

In the first stage, we applied an EMOS method, 'dressing' the output from the $m$ ensemble member forecasts, $x_1, \ldots, x_m$ using a parametric probability density function (pdf) of the following general form:

$$y | \mu, \sigma^2 \sim f\left(\mu, \sigma^2\right) \tag{1}$$

Here $y$ is the concentration of the chemical pollutant, and $\mu$ and $\sigma^2$ are the expected mean and variance of the pdf, $f$, respectively. The expected mean and variance are estimated from the ensemble member forecasts

$$\begin{cases} \mu = b_0 + b_1 x_1 + \ldots + b_m x_m & \text{(2a)} \\ \sigma^2 = c + dS^2 & \text{(2b)} \end{cases}$$

The equation in (2a) encodes for a bias-corrected linear combination, with regression coefficients $b_0, \ldots, b_m$ reflecting the overall performance of the ensemble members over the training period relative to the other members. Equation (2b) implements the so-called spread-skill relationship (Whitaker and Loughe, 1998), with a non-homogeneous variance that depends linearly on the ensemble variance, $S^2 = \frac{1}{m} \sum_{k=1}^{m} (x_k - x^*)^2$, where $x^* = \frac{1}{m} \sum_{k=1}^{m} x_k$ denotes the ensemble mean. This formulation allows the predictive distribution to exhibit more uncertainty when the ensemble dispersion is large, and less uncertainty when the ensemble dispersion is small.

We estimated the coefficients in (2) using a 'global' approach, i.e. a single global calibration was trained across all data using the observations from the last $N$ days to predict the concentration for the upcoming day (Bertrand et al., 2022). This process was repeatedly applied for every day, mimicking an operational forecasting system, using the previous three days to train the algorithm. With a global approach, and with the use of such a short training window, meteorological perturbations on synoptic




scales, or changes in emission strengths, can be quickly accounted for through the variation of the parameters estimated during
the calibration phase.

We exploited the *crps* (*continuous ranked probability score*) (Gneiting et al., 2007) to optimise the values of coefficients, and
applied diagnostic tools, such as the PIT histogram, to evaluate the performance of the calibration stage. The *crps* combines
calibration and accuracy in one index, thus allowing the evaluation of predictive performance, based on the paradigm of
accuracy maximisation subject to calibration (Gneiting et al., 2007). Full details about this procedure are given in A.

**3.2 Stage 2: The statistical modelling of the space-time process**

For a given well-calibrated ensemble prediction, we can exploit additional information allowing higher predictive power
(Chang et al., 2020; Singh et al., 2013; Xi et al., 2015). To this aim, we combined the advantages of well-calibrated ensemble
results with ancillary predictors, to construct a final spatio-temporally resolved model, which will potentially outperform even
the calibrated predictions.

Similarly to other studies (Blangiardo et al., 2013; Cameletti et al., 2013; Fioravanti et al., 2021), for a given calibrated
ensemble prediction, $y(t, s_i)$ at time $t$ and spatial location $s_i$, we assumed the following model:

$$y(t, s_i) = \alpha + \mathbf{z}(t, s_i)\boldsymbol{\beta} + \xi(t, s_i) + \epsilon(t, s_i) \tag{3}$$

Here, $\alpha$ represents the overall, space and time constant, average; $\mathbf{z}(t, s_i) = (z_1, \ldots, z_p)$ the vector of $p$ spatio-temporal predic-
tors, each estimated at the same time, $t$, and spatial location, $s_i$, of the calibrated ensemble prediction, and $\boldsymbol{\beta} = (\beta_1, \ldots, \beta_p)$
the corresponding coefficients vector; $\xi(t, s_i)$ encodes for the residual space-time correlation ones the large-scale component
$\mathbf{z}(t, s_i)\boldsymbol{\beta}$ is accounted for, and $\epsilon(t, s_i)$ the residual unexplained error, assumed to be generated by a Gaussian white noise pro-
cess independent over space and time. We exploited the `r-inla` package (Bakka et al., 2018) to perform all computations for
this second stage exploiting the INLA-SPDE approach. The details about the parameterisation for each component in (3) are
given in B.

**3.3 Validation**

In order to evaluate the improvement of the predictive qualities of the results of both the first and second stage, we followed
a cross-validation approach, splitting the monitoring stations in two dataset: 668 monitoring stations ($\approx 90\%$) were used to
train the model in the first stage and then fit the INLA-SPDE model; the remaining 82 ($\approx 10\%$) for validation purposes. As
already outlined, the monitoring stations are not evenly distributed among the different area type; to mitigate this uneven
representativeness issue and improve fairness during the validation stage, the number of urban, suburban and rural stations was
selected at random in proportion to their number; precisely, 38 (5.1%) urban, 21 (2.8%) suburban and 23 (3.1%) rural stations
were selected for validation purposes and the remaining part was left for training.

A second level of validation was also applied in 'forecasting mode': both the output from the first or second stage can be
used to predict the concentration for the next day, i.e. both the parameters estimated during the first stage or the INLA-corrected





values from the second stage can be used to predict the concentrations for the next day, mimicking what could happen when the post-processing phases are applied in a true time forecasting mode.

We assessed the performance of the post-processing stages, using well-known and widely used scoring indexes: root mean square error, bias, correlation coefficient and contingency tables. Moreover, the PIT histograms and credible intervals were used to assess accuracy and reliability.

The contingency tables were built using the thresholds defined by the current Italian legislation (borrowed from the European one) and the new guidelines indicated by the World Health Organisation (WHO), which has reviewed the most recent epidemiological evidences. WHO set stringent and challenging short-term guideline levels and interim targets (WHO, 2021); for example, the current threshold value from the Italian legislation for daily $PM_{10}$ concentration is 50 $\mu g/m^3$, 120 $\mu g/m^3$ for the maximum 8-hour daily value for ozone, and 200 $\mu g/m^3$ for the maximum hourly value for $NO_2$. The new WHO air quality guidelines are equal to 45 $\mu g/m^3$ for daily $PM_{10}$, 15 $\mu g/m^3$ for daily $PM_{2.5}$, 100 $\mu g/m^3$ for the maximum 8-hour daily value for $O_3$, and 25 $\mu g/m^3$ for daily $NO_2$ concentration.

## 4 Results

### 4.1 Exploratory analysis

In order to assess the value of raw CAMS air quality forecasts, here we introduce the same approach described in Murphy (1988), based on the use of a skill score, i.e. a measure of the accuracy of the forecast relative to the accuracy of forecast produced by a *standard of reference*. Precisely, we measure the added value by means of the skill score, $SS$, defined as:

$$SS = 1 - \frac{RMSE_f}{RMSE_r} \qquad (4)$$

where $RMSE_f$ is the root mean square error of forecasts, and $RMSE_r$ is the root mean square error of the reference used as no-skill baseline. The previous day's observations are used as reference baseline; in this case the skill score measure the accuracy of the CAMS forecast in predicting the next day value compared to the hypothesis of persistence, i.e. that concentration does not change from the previous day. Note that $SS$ is positive when the accuracy of the forecast is greater than the accuracy of the reference baseline, and the added value becomes more and more important as the skill score approaches one. Moreover, negative values of the skill score mean that, on average, the performance of the persistence hypothesis overcomes that of the raw CAMS forecast.

The results are reported in Figure 1 where the CAMS results for the next day prediction against persistence are evaluated in terms of the skill score defined in (4). On the average the root mean square error is about 12 $\mu g/m^3$ for the daily mean $PM_{10}$ concentration, 9 $\mu g/m^3$ for $PM_{2.5}$, 28 $\mu g/m^3$ for the 1-hour $NO_2$ daily maximum and 21 $\mu g/m^3$ for the $O_3$ highest 8-hour daily maximum, but the persistence-based forecast (from the observed previous day values) performs consistently better than the model-derived values, so that the skill score is systematically negative for all models and pollutants. In particular, for the 1-hour $NO_2$ daily maximum, the persistence-based prediction allows to almost halve the error in the next day prediction for almost all models, indicating the problems they have in predicting the concentration peaks on a small time scale, probably due



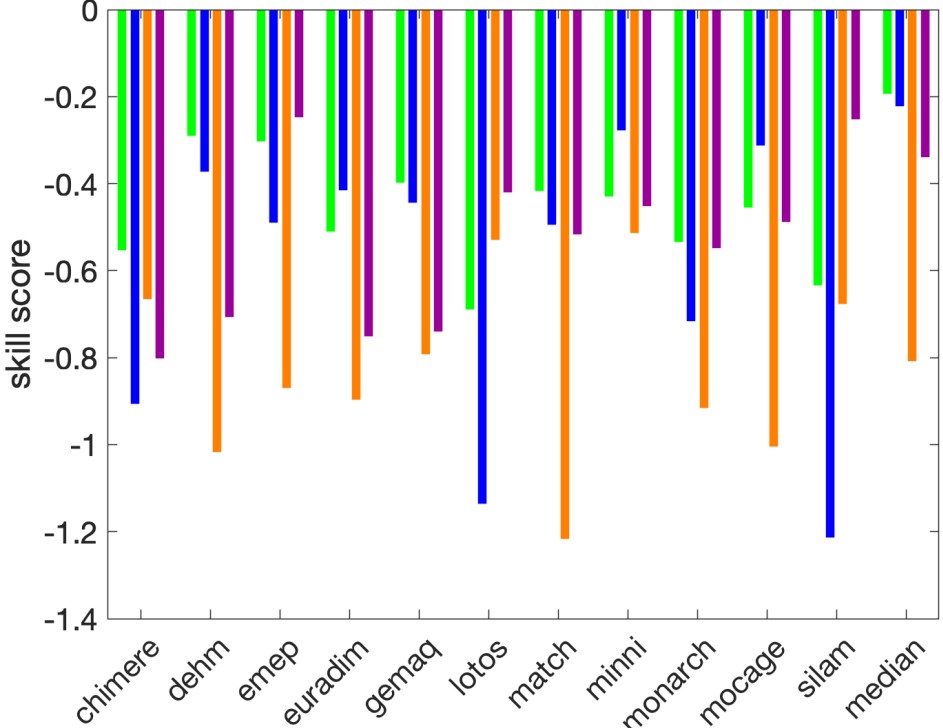

**Figure 1.** Skill score for the CAMS models. For each model the skill score is reported for the 24-hour look-ahead forecast during the year 2022 compared to the prediction based on the persistence of the previous day concentration for $PM_{10}$ (green), $PM_{2.5}$ (blue), $NO_2$ (orange) and $O_3$ (purple bars)

to the low spatial resolution. Also note that in some cases the skill score is even lower than -1, meaning that the root mean square error of the raw CAMS predictions is more than double that obtained by exploiting the persistence assumption. The median model is only partially able to remedy this condition, usually showing an improvement over the prediction made by the individual models, but with a still negative skill score. Even if we disentangle results among the different area type monitoring station (data not shown), the same general conclusions about the skill of the raw CAMS predictions still continue to be valid.

The first conclusion we can draw from this preliminary analysis is that the CAMS ensemble results need to be re-calibrated in order to remove the bias and improve accuracy. This is what is described in the next subsections.





## 4.2 The added value of the post-processing stages: deterministic-style assessment

### 4.2.1 Root mean square error, bias and correlation

We now give the results of applying the first and second post-processing stage to the next day predictions for $PM_{10}$, $PM_{2.5}$, $NO_2$ and $O_3$. First, we assessed the performance of the post-processing stages in terms of deterministic scores. Table 2 provides a summary of some of well-known and widely used scoring measures, i.e. root mean square error, bias and correlation. The same information is also reported in graphical form in Figure 2, i.e. as Taylor diagrams, for the validation and prediction dataset.

As expected, the RMSE (root mean square error) and the bias are strongly reduced for the training dataset for every pollutant. For example, for $PM_{10}$, the RMSE is reduced by more than half, but the same is also valid for all other pollutants. As can be seen, ensemble raw data for $PM_{10}$, $PM_{2.5}$ and $NO_2$ are affected by a negative bias, which is almost zeroed after the application of both the first and second post-processing stage. The high values of the correlation coefficients for the training set (above 0.75 for $PM_{10}$, $PM_{2.5}$ ad $O_3$ after the first stage, and above 0.85 after the second stage) show that the predicted and the observed values are well in accordance. Lower scores are obtained for $NO_2$, for which only the exploitation of auxiliary spatio-temporal predictors (in the second stage) is able to raise its value up to 0.85.

However, it is clear that the results obtained for the training dataset are not suitable for a fair comparison. A more reliable estimate of the performance of the post-processing stages can be obtained from the validation dataset. These data represent 10% of the measurement stations, randomly selected but stratified in proportion to the type of area in which they are located. The validation dataset has not been included in the training process, so that the results from the validation dataset can be considered as a more reliable and truthful estimation on the model performance at different spatial locations. In the case of the validation dataset, we still have a strong reduction of the RMSE and the almost zeroing of the average bias, and a consistent high correlation (usually greater than 0.80), especially after the second stage.

The prediction dataset refers to the same monitoring stations used for training, but the post-processing framework is exploited to predict the next day concentrations. As expected, the performances are lower in this case, even if both the first and the second stage have generally introduced significant improvements both in terms of RMSE, bias and correlation.

The Taylor diagrams in Figure 2 report the same information in a graphical compact format.

### 4.2.2 Sensitivity, specificity and threat score

In order to assess the ability of raw CAMS data, or post-processing models, to predict the exceeding of a given threshold, we built a confusion matrix, categorising each prediction into a true/false positive/negative outcome. The counts from the confusion matrix were used to define the following indexes: 1) *sensitivity*, also known as 'true positive rate', defined as the ratio between the number of true positives to the total number of observed exceedances; 2) *specificity*, also known as 'true negative rate', defined as the ratio between true negatives to the total number of observations not exceeding a given threshold; 3) *threat score*, also known as 'critical success index' or 'Jaccard index', defined as the ratio between the number of true positives to the total number of predicted or observed exceedances.





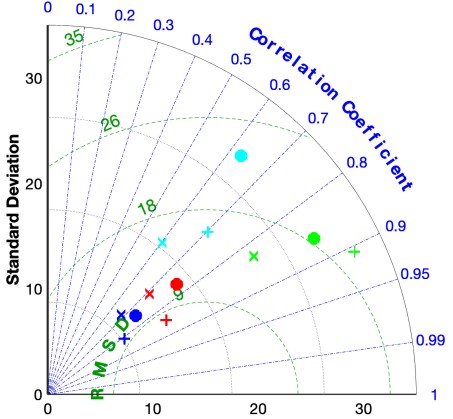 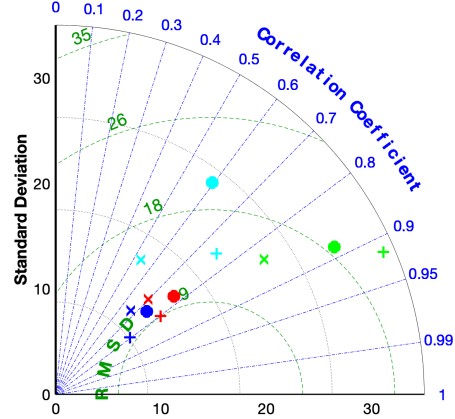

**Figure 2.** Taylor diagrams for the validation (left) and prediction (right) dataset. For each figure, the red symbols refer to $PM_{10}$; blue to $PM_{2.5}$; cyan to $NO_2$ and green to $O_3$. For each pollutant, the 'x' markers refer to the performance of the raw CAMS data; circles to the results after the first post-processing stage, and '+' after the second post-processing stage.

We can consider sensitivity as a measure of how well our predictions can correctly identify exceedances and specificity as a measure of how well our predictions can correctly identify when observations fall short of a given threshold, while the threat score can be seen as a measure of the overlap between the distribution of observations versus that of predictions. A perfect forecast would take a value of 1 for all these indexes.

The sensitivity, specificity and threat score indexes are plotted in Figures (3) and (4) for the validation and prediction dataset, respectively, where the number of exceedances were defined with respect to the threshold from the new WHO guidelines. For $PM_{10}$ and $NO_2$, raw CAMS data show a low precision ($\approx 0.4$), which is greatly improved after the first and second post-processing stage, achieving a value as high as (or even higher than) 0.8. This means that most events above the threshold are missed from the raw CAMS data, but almost always as expected after post-processing stages.

The increase in sensitivity is not accompanied by a decrease in specificity; in most cases, on the contrary, post-processing increases specificity, i.e. the number of events correctly classified as below the threshold. The only exception is represented by $NO_2$, for which the specificity decreases after the post-processing stages. However, it should also be said that 25 $\mu g/m^3$ represents a very low threshold for the 1-hour daily maximum, therefore a low specificity in capturing events at such low concentrations is expected.

**4.3    The added value of the post-processing stages: probabilistic-style assessment**

RMSE, bias and correlation look for a matching between observations and training/validation/prediction dataset in a 'stiff' mode. However, both the first and the second post-processing stages tailor a statistical 'dress' around results, so that we can use probabilities in measuring the properties of our approach.





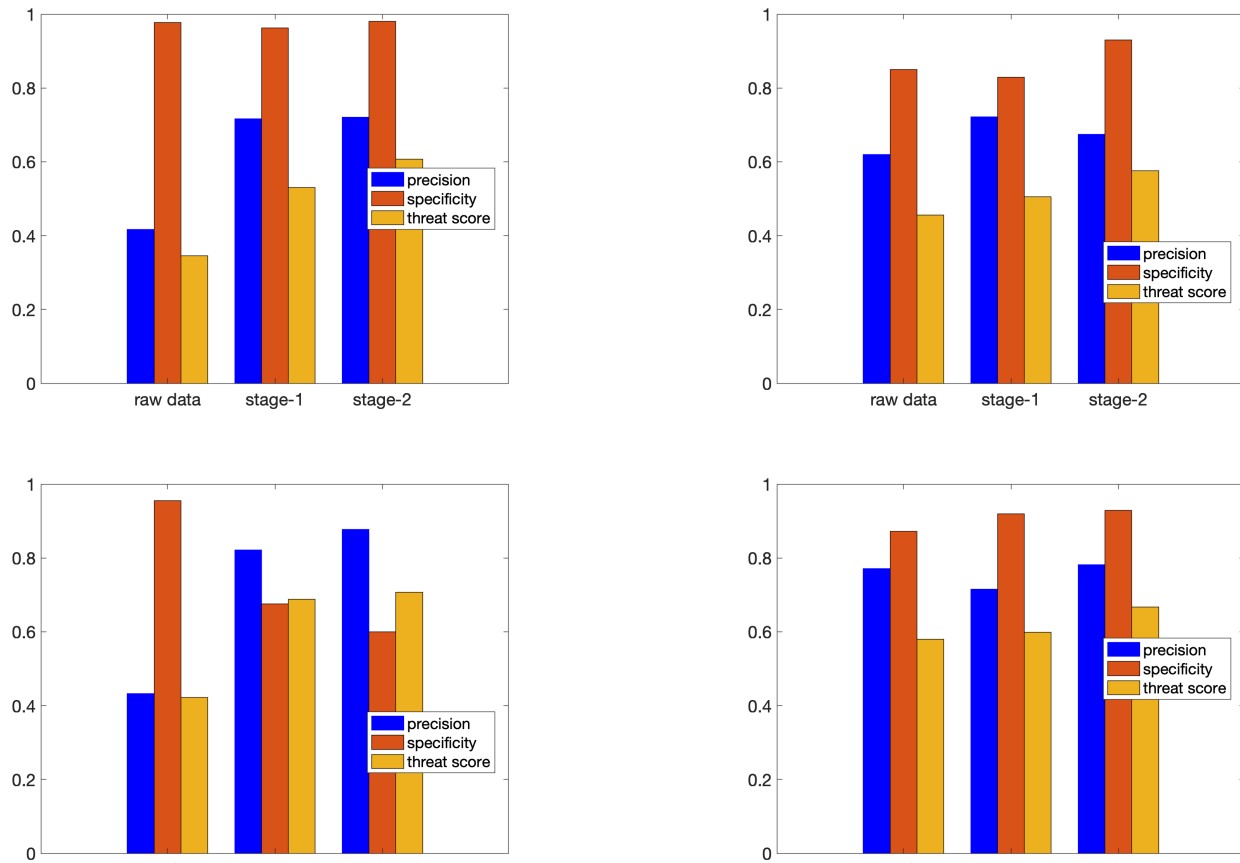

**Figure 3.** Scores (sensitivity, specificity and threat score) for the validation dataset for $PM_{10}$ (upper-left panel), $PM_{2.5}$ (upper-right panel), $NO_2$ (lower-left panel) and $O_3$ (lower-right panel). The blue bars correspond to the raw CAMS results, while the results after the application of the first and second stage are reported as orange and yellow bars, respectively. The number of exceedances (for both observations and predictions) are defined with respect to the new WHO guidelines: 45 $\mu g/m^3$ for daily $PM_{10}$, 15 $\mu g/m^3$ for daily $PM_{2.5}$, 100 $\mu g/m^3$ for the maximum 8-hour daily value for $O_3$, and 25 $\mu g/m^3$ for daily $NO_2$ concentration.

### 4.3.1 Reliability and accuracy

First, we checked whether our approach ensures reliability while maintaining high accuracy. In a meteorological context, reliability measures the capability of unbiased predictions to closely follow the observed frequencies, i.e. for a perfectly reliable forecast, an event declared to occur with frequency $p$ is actually forecast with a proportion $p$ on average (Taylor, 2001). Instead, accuracy refers to how close the prediction is to the observed data. Both are concerned with the conditional probability to predict an observation for a given forecast. An in-depth discussion of those and other attributes of probabilistic forecasts can be found

in Jolliffe and Stephenson (2011).





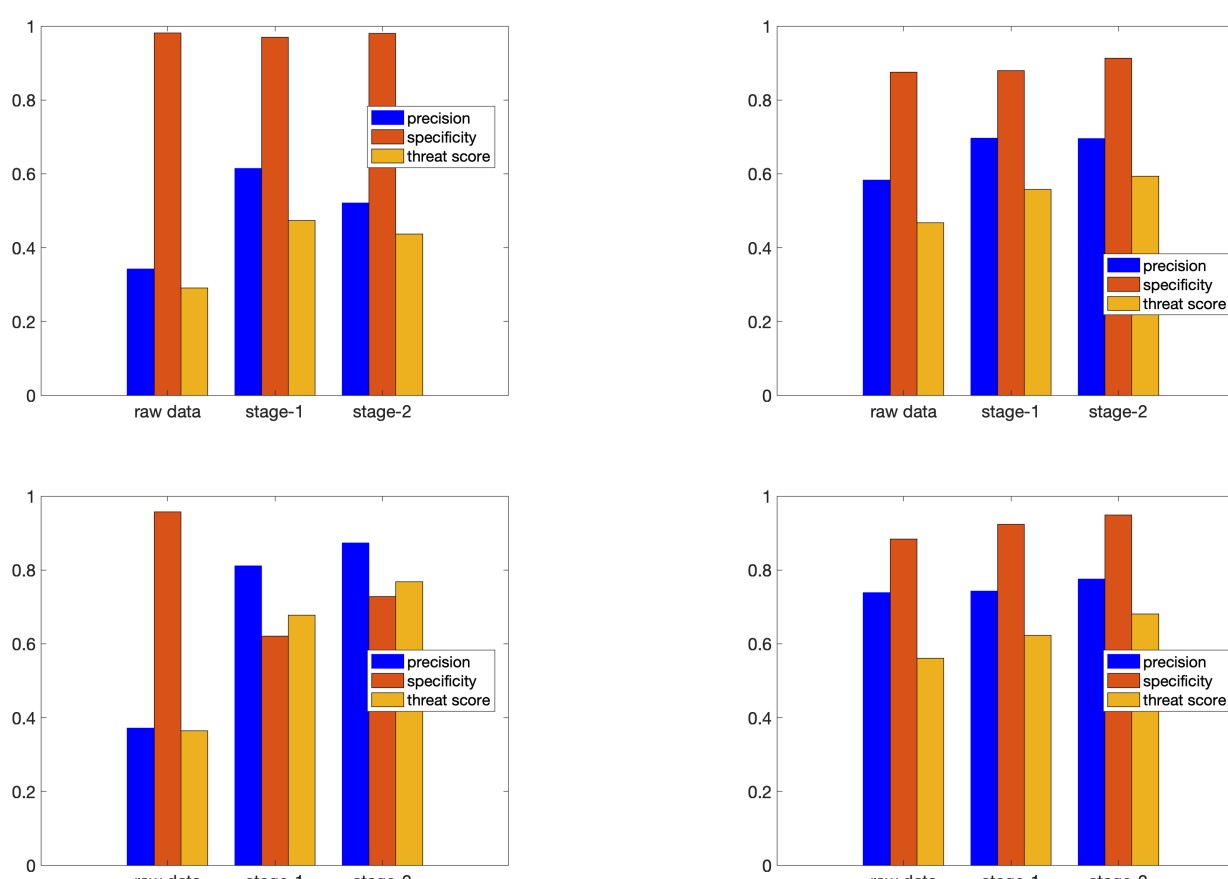

**Figure 4.** Same as in Figure 3 but for the prediction dataset.

Gneiting et al. (2005), in their seminal work, stated that the goal of a well calibrated probabilistic forecast is to maximise accuracy, subject to reliability. Figure 5 shows the probability integral transform (PIT) for the raw CAMS predictions and after the application of the first and second post-processing stage to the validation dataset. Figure 6 shows the same results, but for the prediction dataset. As can be seen, the PIT histograms for the raw CAMS results for $PM_{10}$ and $NO_2$ follow a quasi-
305 monotonic decreasing trend, meaning that raw CAMS results tend to underestimate observations, while the PIT histogram for $O_3$ shows an inverted-U shape profile, meaning overdispersive behaviour, i.e. unnecessarily wide prediction intervals that have higher than nominal coverage. Conversely, the histograms for the validation and prediction dataset, after applying the first and second stage, are closer to a flat profile, showing a more accurate reproduction of the probabilities of occurrence, tending to mitigate both the overall bias and over/under-dispersion effects.





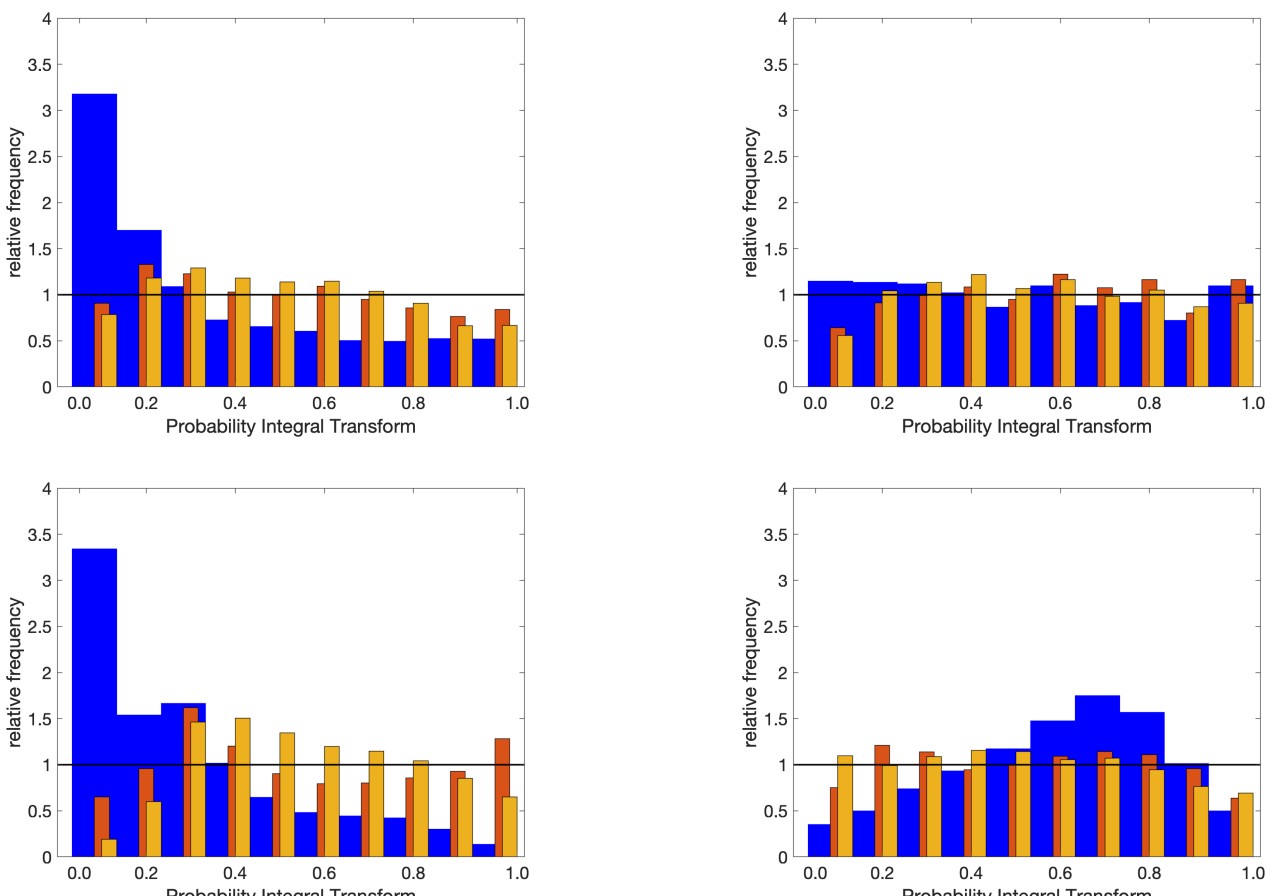

**Figure 5.** PIT for $PM_{10}$ (upper-left panel), $PM_{2.5}$ (upper-right panel), $NO_2$ (lower-left panel) and $O_3$ (lower-right panel). The blue bars correspond to the raw CAMS results, while the results after the application of the first and second stage to the validation dataset are reported as orange and yellow bars, respectively. The orange and yellow bars have been slightly shifted and re-sized in width, so as not to completely overlap the blue bars. The black horizontal lines have been drawn for reference: for a perfect reliable ensemble the PIT should be flat, at a relative frequency equal to 1.

### 4.3.2 Credible intervals

The construction of credible intervals from the cumulative distribution function (cdf) is straightforward. For instance, the 25th and 75th percentile of cdf form the lower and upper endpoints of the 50% central prediction interval, respectively, from which the sharpness, i.e. the spread around the predicted value, can be evaluated. For a well-calibrated ensemble, the higher the accuracy, the more data is concentrated around the predicted value, the more value the model adds.

We estimated the 25th and 75th percentile from the posterior distributions of the first and second stage for each pollutant, and compared these results with the interval from the 25th to the 75th percentile from the raw CAMS ensemble data. Table 3 shows





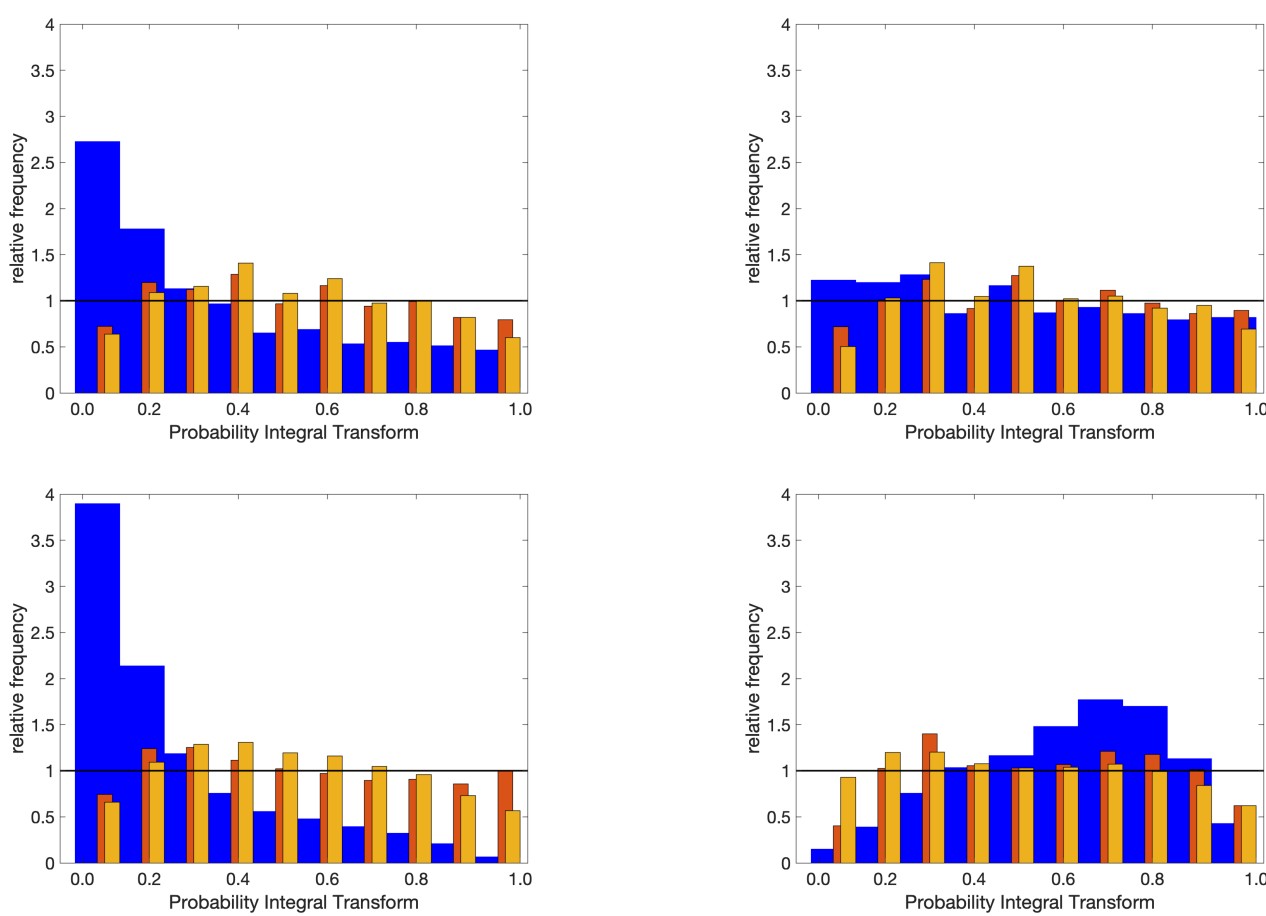

**Figure 6.** Same as for Figure 5, but for the prediction dataset.

the average widths of the 50% probability interval for the raw CAMS data and after the application of the first and second post-processing stage. As can be observed from this table, after the application of the first stage, the credibility interval tends to widen, i.e. the calibrated data show a much smaller bias (see Table 2) but at the cost of widening the credibility interval, making the prediction less accurate. On the other hand, the effect of applying the second stage, through the exploitation of spatial and spatio-temporal predictors, is not only to improve the accuracy of the forecast, but also to make the forecast sharper, narrowing the credibility interval. This range is also generally smaller than that obtained from the raw CAMS data. For example, the credibility interval for all pollutants is roughly halved for the validation dataset, going from 8.3 to 4.4 $\mu$g/m$^3$ for PM$_{10}$, from 8.3 to 4.4 $\mu$g/m$^3$ for PM$_{10}$, from 5.3 to 3.3 $\mu$g/m$^3$ for PM$_{2.5}$, from 17.9 to 10.8 $\mu$g/m$^3$ for NO$_2$, and from 13.5 to 15.1 $\mu$g/m$^3$ for O$_3$.



## 4.4 Examples of application

Finally, we want to conclude this section with two examples of potential applications of our post-processing analysis: 1) real-time time series forecasting and 2) interpolation at high spatial resolution for the detection of non-compliant areas. We provide some information about these two application examples using data for $PM_{10}$.

Figure 7 shows the predictions made for $PM_{10}$ for some monitoring stations, illustrative of rural, urban and sub-urban area types. The second-stage predictions are shown, exploiting the spatial and spatio-temporal ancillary information to correct the bias of CAMS models. These predictions represent the next day's forecast, i.e. at a 24-hour look-ahead time window, in an attempt to simulate the processes that should be activated if concentrations exceed the warning threshold. As expected, the $PM_{10}$ concentrations for the remote station are quite low, especially for the summer period, slightly exceeding 20 $\mu g/m^3$ in

the most extreme conditions. In the urban station (C.so Francia, Rome), on the other hand, the observed concentrations are higher (even higher than 60 $\mu g/m^3$ for the selected station) in the winter period. This is a well-known phenomena, linked to the convective processes transporting particles emitted at low levels to higher altitudes in the summer period; conversely, urban areas are affected by higher concentrations of particulate matter in the summer period, due to condensation phenomena at low temperatures and atmospheric subsidence (Marinoni et al., 2008).

The forecasts from the CAMS ensemble raw data predict relatively higher concentrations than observed in the winter period, and relatively lower concentrations in the summer period for the rural stations, and are consistently lower than observed for the urban station in the winter period. The suburban station was selected as an example of a site for which the raw CAMS predictions are in good agreement with experimental data. In any case, the corrections made by the post-processing mechanism allow the predictions to align with the observed profiles, to a level compatible with the displayed credibility intervals, and even

improve the agreement with data from the suburban station. These graphs suggest that the model is able to remove much of the systematic bias, predict the temporal variability in the monitoring sites and provide accurate and reliable predictions.

A second application concerns the possibility of extrapolating the forecasts obtained at the monitoring sites to other spatial locations as well. To this end, a regular spatial grid for Italy has been set up, at a resolution of $4 \times 4$ km$^2$, and the spatial and spatio-temporal predictors have been reconstructed for each of these grid points. In this way, it was possible to highlight the

350 non-compliant areas with a spatial resolution higher than that made available by the CAMS models and with greater accuracy. Figure 8 shows the map of the 90.4th percentile of daily means for $PM_{10}$. Similar maps can be obtained also for the other pollutants. Red and purple marked areas of this map indicate values above 50 $\mu g/m^3$, indicating where this threshold has been predicted to be exceeded for more than 35 days in the calendar year. Not surprisingly, large areas with concentrations above the daily limit value are observed in northern Italy (i.e. the Po Valley) and in the main urban and surrounding areas, where people

are often exposed to average levels above 50 $\mu g/m^3$. Accurate prediction of these exceedances, and of the areas in which they occur, could be used to activate more effective prevention measures.



## 5 Conclusions

In this work, the effectiveness of statistical post-processing techniques aimed at improving the accuracy and reliability of the predictions of the air quality models of the CAMS suite have been tested. It is well known that the CAMS suite (currently made up of eleven members), while representing the state of the art of atmospheric modelling, show significant biases, for which it is advisable to adopt post-processing techniques that are statistically reliable and computationally inexpensive to cope with operational constraints. Furthermore, predictions are still available at a moderate spatial resolution $(0.1° \times 0.1°)$, and may miss the steep spatial gradients that occur in the vicinity of large urban areas and industrial sites.

In order to ameliorate these problems, a statistical post-processing technique was developed and applied to the Italian region, capable of correcting both the bias and the reliability of ensemble predictions. The concentrations of the main air pollutants, $PM_{10}$, $PM_{2.5}$, $NO_2$ and $O_3$, were taken into consideration, and a new two-stage post-processing approach, able to fulfill the operational constraints, was designed. In the first stage, ensemble data were combined together through the minimisation of the continuous ranked probability score (crps) over the training data. During the second stage, the ensemble prediction was corrected exploiting additional spatio-temporal predictors within a framework based on the INLA-SPDE approach. The post-processing stages make use of a short training period (three days), so as to rapidly adapt to changes in meteorological or emission conditions, and apply simultaneously to all monitoring stations.

The validation procedure shows that the post-processing stages were able to remove the systematic biases, improve accuracy and provide reliable forecasts. Moreover, the global approach allowed the application of the INLA-SPDE framework to a regularly spaced spatial grid (at a resolution higher than the original CAMS members), highlighting the regions in which exceedances occur. The post-processing correction process has been applied to the measurement stations for the year 2022 for Italy, but this procedure can be easily generalised to any spatial and temporal region. Because of its flexibility, we also expect that this global approach is prone to adapt in real time to fast changes in meteorological conditions and/or abrupt changes in pollutant emissions.

*Code and data availability.* The full list of source codes and dataset used in this work are archived by the authors and can be obtained from the corresponding author upon request

*Sample availability.* The source codes, along with a sample of input files, are available from github, and a local clone can be generated by the command: `git clone https://github.com/angeloriccio/EMOS.git`



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

## Appendix A: The ensemble model output statistical approach

A number of different pdfs have been proposed for $f$ in (1): normal, truncated normal, logistic, gamma and other distributions; the reader is referred to Wilks (2018) for an extensive discussion and comparison. Among all possible choices, after an exploratory phase, we found that an effective approach consists in the selection of the gamma distribution, $\mathcal{G}(\alpha, \beta)$.

The gamma probability distribution function (pdf) is:

$$\mathcal{G}_{\alpha,\beta}(x) = \frac{\beta^\alpha}{\Gamma(\alpha)} x^{\alpha-1} e^{-\beta x} \quad \text{for } x > 0 \quad \alpha, \beta > 0 \tag{A1}$$

characterised by the 'shape' parameter $\alpha$ and the 'inverse scale' parameter $\beta$. $\Gamma(\cdot)$ is the gamma function. The shape and inverse scale parameters are related to the predicted mean, $\mu = \alpha/\beta$, and variance, $\sigma^2 = \alpha/\beta^2$, which in turn are related to the ensemble forecasts, $x_1, \ldots, x_m$, by the relations shown in (2).

Gneiting et al. (2007) proposed to evaluate the coefficients in (2) using a diagnostic approach based on the minimisation of the *continuous ranked probability score* (*crps*). The *crps* is the integral of the Brier scores at all possible threshold values $t$ for the continuous predictand (Toth et al., 2003). In plain terms, the *crps* is defined as:

$$crps(F_{\alpha,\beta}, y) = \int\limits_{-\infty}^{\infty} \left[ F_{\alpha,\beta}(t) - H(t-y) \right]^2 dt \tag{A2}$$

where $H(t-y)$ is the Heaviside function and takes the value 0 when $t < y$ and 1 otherwise, and $F_{\alpha,\beta}$ is the cumulative 525 distribution function (cdf) corresponding to the pdf in (A1). A closed form of the *crps* for the gamma pdf has been obtained





by Scheuerer and Möller (2015), making the minimisation procedure easy and fast. For an observation-forecast pair $(y, \mathbf{x})$, the *crps* closed form reads:

$$crps(y, \mathbf{x}) = y \left(2F_{\alpha,\beta}(y) - 1\right) - \frac{\alpha}{\beta} \left(2F_{\alpha+1,\beta}(y) - 1\right) - \frac{1}{\beta \mathcal{B}\left(\frac{1}{2}, \alpha\right)} \tag{A3}$$

with $y$ being the observation, and $\mathcal{B}$ the beta function. The forecast vector $\mathbf{x} = (x_1, \ldots, x_m)$ enters into (A3) via the shape and
inverse scale parameters. Their expressions in terms of the expected mean and variance read $\alpha = \mu^2/\sigma^2$ and $\beta = \mu/\sigma^2$. The mean and variance, in turn, depend on the coefficients used in (2). In case of a training set of observations and forecasts, the quantity to be minimised is

$$crps = \frac{1}{N} \sum_{i=1}^{N} crps(y_i, \mathbf{x}_i) \tag{A4}$$

with $i$ denoting the $i$th observation-forecast pair and $N$ is the total number of pairs in the training set.
As also implemented in Gneiting et al. (2005), we strengthened the estimate of the coefficients in (2a), in order to avoid negative values, which can be caused by collinearities among the members of the ensemble.

   In order to estimate the expected mean and variance from the *crps* minimisation in (A4), we are left with the selection of the length of the training period. This aspect was already faced in Bertrand et al. (2022) and Gneiting et al. (2005), where different sliding windows, from 3 to 62 days, were considered. Of course, there is a trade-off in selecting a specific training length: a
longer training period reduces the statistical variability in the estimation of coefficients; a shorter training period is able to adapt more rapidly to different conditions, e.g. meteorological perturbations or changes in the emission scenarios. In our case, we found that even a very short training period is able to achieve good performance. Results shown in this work refer to a sliding training period of three days, i.e. all predictions for the next day were obtained using air quality and meteorological data from the previous three days. For each day, this process was repeatedly applied, mimicking an operational forecasting system.

**Appendix B: The spatio-temporal statistical model**

Results from the first stage are used to feed a second stage, in which we introduce additional spatio-temporal predictors. For a given calibrated ensemble prediction, $y(t, s_i)$, at time $t$ and spatial location $s_i$, we assumed the model shown in (3). In this case, we model the residual as a first-order autoregressive model with spatially correlated innovations, $\omega(t, s_i)$:

$$\xi(t, s_i) = a\xi(t-1, s_i) + \omega(t, s_i) \tag{B1}$$

for $t = 2, \ldots, T$ and $|a| < 1$, $\xi(t, s_i)$ derives from the stationary distribution $\xi \sim \mathcal{N}(0, \sigma_\omega^2/(1-a^2))$, where $\mathcal{N}(\eta, \varepsilon^2)$ denotes the Gaussian distribution with mean $\eta$ and variance $\varepsilon^2$. Moreover, $\omega(t, s_i)$ is assumed to be temporally independent and characterised by the spatio-temporal covariance function

$$\mathrm{Cov}\left(\omega(t, s_i), \omega(t', s_j)\right) = \begin{cases} 0 & \text{if } t \neq t' \\ \sigma^2 \mathcal{C}(h) & \text{if } t = t' \end{cases} \tag{B2}$$



for $i \neq j$. The purely spatial correlation function $\mathcal{C}(h)$ depends on the spatial location $s_i$ and $s_j$ only through the Euclidean
distance $h = \parallel s_i - s_j \parallel \in \mathcal{R}$; thus, the process is, according to the nomenclature used in the geostatistical literature, a second-order stationary and isotropic process (Cressie and Wikle, 2015). For the specification for the purely spatial covariance function, $\mathcal{C}(h)$, we follow the common choice of the Matérn function

$$\mathcal{C}(h) = \frac{1}{\Gamma(\nu)2^{\nu-1}}(kh)^{\nu} K_{\nu}(kh) \tag{B3}$$

with $K_{\nu}$ denoting the modified Bessel function of the second kind and order $\nu = 1$. The parameter $\nu$ measures the degree of smoothness of the process; instead, $k > 0$ is a scaling parameter related to the range $\rho$, i.e. the distance at which the spatial correlation becomes small. In particular, we use the empirically derived definition $\rho = \sqrt{8\nu/k}$, with $\rho$ corresponding to the distance where the spatial correlation is close to $0.1$ (Lindgren et al., 2011). This kind of model is well discussed and widely used in the air quality literature, thanks to its flexibility in modelling the effect of relevant predictors as well as space and time dependence (Blangiardo et al., 2013; Cameletti et al., 2013; Fioravanti et al., 2021; Konstantinoudis et al., 2022).

For the second stage a training period of three days was chosen, too, and the estimation process was repeated for each day.

*Author contributions.* AR worked on the implementation of the study and performed the simulations with support from EC. EC was responsible for the acquisition of the observed air quality data. AR performed the analysis with the support of EC for results interpretation. AR wrote this article, with contributions from EC.

*Competing interests.* The authors declare that they have no conflict of interest





**Table 2.** Statistics of the cross-validation study. RMSE is the root mean-square error; CC is the correlation coefficient. Units for the RMSE and the bias are expressed in $\mu g/m^3$ for all pollutants.

| | | PM$_{10}$ | | | PM$_{2.5}$ | | | NO$_2$ | | | O$_3$ | | |
| --- | --- | --- | --- | --- | --- | --- | --- | --- | --- | --- | --- | --- | --- |
| | | CAMS data | stage 1 | stage 2 | CAMS data | stage 1 | stage 2 | CAMS data | stage 1 | stage 2 | CAMS data | stage 1 | stage 2 |
| RMSE | training | 12.32 | 9.99 | 5.13 | 8.80 | 8.22 | 4.06 | 28.43 | 22.34 | 13.02 | 21.62 | 16.18 | 7.42 |
| | validation | 12.21 | 10.82 | 7.96 | 7.91 | 7.71 | 5.74 | 26.59 | 24.36 | 19.02 | 19.97 | 16.51 | 13.99 |
| | prediction | 12.26 | 9.92 | 9.09 | 8.74 | 8.15 | 11.82 | 28.43 | 22.36 | 16.40 | 21.61 | 16.18 | 14.14 |
| Bias | training | -5.86 | 0.22 | -0.63 | -0.88 | 0.39 | -0.52 | -19.36 | 0.99 | -2.21 | 8.44 | 0.35 | -0.77 |
| | validation | -5.47 | 0.86 | -0.04 | 0.29 | 1.67 | -0.54 | -16.29 | 4.30 | 1.49 | 7.03 | -1.30 | -0.92 |
| | prediction | -5.86 | 0.20 | -0.97 | -0.87 | 0.37 | -0.58 | -19.35 | 0.98 | -2.35 | 8.43 | 0.37 | -1.54 |
| CC | training | 0.70 | 0.77 | 0.94 | 0.67 | 0.74 | 0.93 | 0.54 | 0.59 | 0.85 | 0.84 | 0.88 | 0.98 |
| | validation | 0.71 | 0.76 | 0.85 | 0.68 | 0.75 | 0.81 | 0.60 | 0.63 | 0.70 | 0.83 | 0.86 | 0.91 |
| | prediction | 0.70 | 0.77 | 0.79 | 0.67 | 0.74 | 0.53 | 0.54 | 0.59 | 0.75 | 0.84 | 0.88 | 0.92 |



**Figure 7.** Prediction of $PM_{10}$ daily concentrations for three monitoring sites. Top panel: rural (Trentino regional network, Monte Gaza). Middle panel: urban (Lazio regional network, C.so Francia, Rome). Bottom panel: suburban (Sicilian regional network, industrial district close to the Agrigento town). Observed values (black dots) are reported versus raw CAMS data (red line) and EMOS post-processed data after the second stage (dashed black line) during the winter (January 2022, left column) and summer (July 2022, right column) season. The yellow filled area represents the interval between the 25th and the 75th percentile, while the grey filled area the interval between the 2.5th and the 97.5th percentile.





**Figure 8.** $PM_{10}$ concentration map of 90.4th percentile of daily means in 2022, after the application of the second post-processing stage and extrapolated over a regular $4 \times 4$ km grid resolution. Values above $50$ $\mu$g/m$^3$ indicate those areas where this threshold has been predicted to be exceeded for more than 35 days in this calendar year.



**Table 3.** Average width for the 50% probability interval around the predicted value for the estimation dataset (first row), validation dataset (second row) and in prediction mode (third row). Units are expressed in $\mu g/m^3$ for all pollutants.

| | PM$_{10}$ | | | PM$_{2.5}$ | | | NO$_2$ | | | O$_3$ | | |
| --- | --- | --- | --- | --- | --- | --- | --- | --- | --- | --- | --- | --- |
| | CAMS data | stage 1 | stage 2 | CAMS data | stage 1 | stage 2 | CAMS data | stage 1 | stage 2 | CAMS data | stage 1 | stage 2 |
| estimation | 8.3 | 11.1 | 3.3 | 5.9 | 8.1 | 2.5 | 19.2 | 25.7 | 7.5 | 14.6 | 23.4 | 7.9 |
| validation | 8.2 | 11.5 | 4.4 | 5.3 | 7.8 | 3.3 | 17.9 | 26.3 | 10.8 | 13.5 | 23.5 | 15.1 |
| prediction | 8.3 | 11.1 | 7.1 | 5.9 | 8.1 | 5.1 | 19.2 | 26.0 | 10.5 | 14.6 | 23.5 | 14.4 |