# Peer review of "Technical Note: Accurate, reliable and high resolution air quality predictions by improving the Copernicus Atmosphere Monitoring Service using machine learning techniques"

_EGUsphere, 2023_

## Author Comment (AC1)

**Reply to referee #2**

1. The title of this manuscript emphasizes the application of machine learning. However, the methods described in this manuscript, such as calibration of the ensemble in the first stage and statistical modelling of the space-time process in the second stage, appear to be statistical methods instead of machine learning.

*We thank the referee for the suggestion and agree with its comment. We suggest changing the title to: "Accurate, reliable and high-resolution air quality predictions by improving the Copernicus Atmosphere Monitoring Service using a novel statistical post-processing method"*

2. calibration of the ensemble: here the objective is to produce calibrated concentrations based on the linear regressions of multi-models and observations with parameters b1, … bm. I am a little surprised to see that this process needs to be repeated every day. I am wondering whether there are temporal and spatial changes in the parameters b1, … bm in the model training and what these changes in the parameters b1, … bm represent.

*For brevity, in the first version of the paper, we did not include details about the b1, … bm properties. However, you raise an important point. We reply to your question under answer '3' (see below)*

3. statistical modelling of the space-time process: similar to the above question, does this process need to be repeated every day?

*The b1, …, bm weights (stage 1) can be interpreted as a measure of the overall performance of each ensemble member, over the training period, relative to the other members. These weights are constructed using monitoring data located throughout the spatial domain. Due to different meteorological conditions and seasonal variabilities, their estimation is repeated regularly using a predefined time window. We tested the training of our global model using the last 3 days, the last 7 days, or the last 14 days, and applied it to predict the concentrations of the next day. We did not find large differences in using different training windows, so we chose the less resource-consuming scenario (a 3-day training period, as described in the paper). A short training window has also the advantage of adapting the bias correction rapidly (in case of rapid changes in meteorological conditions or pollutant emissions, as, for example, experimented during the COVID crisis) and is less computing intensive. The same consideration also applies for the space-time process. This process was repeated every day to mimic an operational system running during the 2022-year period, and a new model was trained regularly with the most recent data to stay close to new forecasting situations.*

*In the revised version, we include a section dedicated to the aspects related to the interpretations that coefficients might have. To give you an idea of the results obtained, we extended our analysis to the whole period from 2020 to 2022 (three years of continuous update of b1, …, bm weights). The following figure (left panel) shows the temporal dependence of these weights for $PM_{10}$ (left panel) and $O_3$ (right panel) for each model. For many models, the weights range from 0.05 to 0.3, reflecting their relative performance. Some models (GEMAQ, MATCH, MOCAGE) show a marked seasonal dependence, with the weight of the GEMAQ model increasing significantly during the summer period, while the weight of the MATCH and MOCAGE models increases during the winter period, indicating both a different performance, depending on the season, and complementarity of these models. It is also interesting to note that for ozone, a pollutant with a marked seasonal cycle, most models perform equally well in both the winter and summer seasons. This analysis is reported in the revised paper (see new Figure 2).*

[Figure]

[Figure]

4. Section 4.2-4.3: I suggest shortening these two sections and perhaps moving some figures into supplement because the parameters in both Stage 1 and Stage 2 are trained with observations, and it is thus expected to see some improvements after these two Stages.

*We partially agree with the comment of the referee. As explained in Section 3.2 (Validation), we split the whole dataset (about 700 monitoring stations) in two sets. ≈ 90% were used to train the model, the remaining set (the validation dataset) was used for validation purposes (not used during the training phase for the first and second stage). Comparison with the validation data set is an integral part of the validation process since it represents a measure of the expected error in forecasting independent data. We consider this information as a significant result of applying our post-processing method. However, we followed your suggestion and shortened Sections 4.2 and 4.3 as much as possible (for example, we discard the old Figure 2, which is a replica of the same information reported in Table 2).*

5. Section 4.4: In contrast, it could be better to extend this section as it is most interesting to the readers. For example, Fig. 7 is not convincing as these three stations may not provide a good representation of the whole domain. It could be better to provide scatter plots to show the overall performances of the predictions; While the high-resolution PM concentrations in Fig. 8 are interesting, it is useful to show the map of the differences between predictions and observations to demonstrate the spatial performance of the predictions; in addition, as the model needs to be trained every day, I am wondering whether the performance of the predictions has seasonal variabilities.

*We expanded the analysis, including:*
  1. *a map for the comparison between observations and model values, highlighting the dependence on the type of monitoring station (urban, suburban and rural) and season (see new Figure 3) and geographical region (see new Figure 4).*
  2. *a comparison extended to all pollutants (not only PM10, see the new section 4.3)*
  3. *box-whiskers plots for the bias of all pollutants (see Figures 3 and 4), equivalent to scatterplots.*

*The statistical prost-processing method works equally well, independently from summer and winter seasons (see new Figure 3). The first stage is able to modulate the seasonal dependency of models (see new Figure 2), and the second stage, starting from the calibrated and more accurate results, improves the forecasts, further reducing the bias (see new Figures 3 and 4)*

6. is it possible that this figure overestimates PM concentrations over rural areas because most stations in Table 1 are higher polluted urban and suburban stations?

*As already stated in the previous answer, we conducted a thorough comparison between observations and model values, highlighting the dependence on the type of monitoring station (urban, suburban and rural). We anticipate that the post-processing method can manage very well the spatial dependence (several spatio-*

*temporal predictors were used precisely because they are able to consider the spatial dependence in the surroundings of the measurement station). In the revised version we include a new figure and a new table (see Table 1 and new Figure 3) showing the distribution of the average bias (distinguished by urban, suburban and rural stations) for all pollutants.*

**Technical Comments:**

A flow chart is suggested to provide a clearer description of the methods.
It would be helpful to provide a List to show the variables which were used in the model training including their temporal and spatial resolutions.

*We inserted a flow chart in the revised version (see new Figure 1). Table A1 includes the temporal and spatial resolutions*

Why the Section number of Stage 1 is 3.1.1 but 3.2 for Stage 2? I assume the description of these two stages is parallel.

*Corrected*

Lines 130-131: why most met predictors are selected at 12 UTC?

*Some met variables, i.e. boundary layer, were selected at 00 and 12 UTC to take in account the daily cycle. The other variables were selected at 12:00 UTC. We considered that day to day variations may be represented by this time level.*

---

## Author Comment (AC2)

**Reply to referee #3**

This paper developed a statistical two stage method to better forecast the pollutant levels. The methods are evaluated with respect to key statistical metrics of both deterministic and probabilistic nature. The idea presented in this work is interesting and is of practical importance. The paper overall has good technical quality, although improvements can be made to further improve the manuscript. I suggest publication of this work after the following comments are addressed.

Major:

Line 173: Why are three days' data used to train the coefficients in the first stage? Are the coefficients sensitive to the number of days used for training.

*The b1, ..., bm weights (stage 1) can be interpreted as a measure of the overall performance of each member of the ensemble, over the training period, relative to the other members. We tested the training of our global model using the last 3 days, the last 7 days, or the last 14 days, and applied it to predict the concentrations of the upcoming day. We did not find significant differences in using different training windows, so that we chose the less resource consuming scenario (a 3-day training period, as described in the paper). Moreover, a short training window also has the advantage to adapt the bias correction rapidly (in case of rapid changes in meteorological conditions or pollutant emissions, as, for example, experimented during the COVID crisis) and is less computing intensive.*

Table 2: From this table it seems that stage 2 worsens the prediction in terms of bias as well the RMSE of PM2.5. What is the reason for this?

*The first stage is a bas correction step, ie. the b1,...,bm parameters are constructed so as to remove the bias The expected result is a very low bias. The second stage introduces new information, i.e. the spatio-temporal covariates. This new information is always beneficial, as shown by the further reduction of the mean squared error (reported in the same table), at the small cost of increasing the mean bias in some cases. In any case, we are talking about rather small deviations. For example, for $PM_{10}$ the bias changes from 0.20 to -0.97 $\mu g/m^3$, which is below 1 $\mu g/m^3$ in both cases. For $PM_{2.5}$ changes from 0.37 to -0.58 $\mu g/m^3$. Similar considerations are also valid for $NO_2$ and $O_3$. Differences are less than one or two orders of magnitude of the typical average concentration values of these pollutants, in line with those reported in other studies. We believe that these errors do not detract from the significance of our statistical treatment, even if at present we cannot exclude a deepening of the nature of this behavior in a future work.*

Line 349: More technical description can be provided, e.g., how are the coefficients used in stage 1 obtained? Are they the same as those trained in previous sections? The application in 4.4 is quite interesting and this section could be expanded to include more details.
Figure 8 and related text: Please provide comparison with observations.
I'd like to see some comments on the computational cost of the current method. Low computational cost indicates sensitivity studies (e.g., with respect to spatiotemporal predictors) can be easily performed to potentially improve the current method.

*The weights estimated in stage 1 were used to obtain an ensemble average, and this average was interpolated onto the new 4x4 km grid (using a bi-linear interpolation). In the second stage, the spatio-temporal predictors are estimated at the cell centers of the 4x4 km grid and then the statistical post-processing is applied. In the revised version, we describe this process more precisely (see section 4.5). Moreover, we expand the analysis of the properties of the post-processing approach, including:*
      *1. a map for the comparison between observations and model values, highlighting the dependence on the type of monitoring station (urban, suburban and rural, see new Figure 3) and geographical regions (stations in north, center and south regions), see new Figure 4).*

2. *a comparison extended to all pollutants (not only PM10, see the new section 4.3)*
3. *scatterplots for all pollutants*

*Computational costs are an important point of our approach. In the literature, the problem of building spatially continuous concentrations maps over large domains has been approached by different perspectives. Most of the studies use hierarchical models based on the Markov chain Monte Carlo (MCMC) approach; despite the existence of user-friendly programming tools, the application of a MCMC approach is rather cumbersome, requiring a lot of CPU-time as well as tweaking of simulation and model parameters' specifications. Some strategies have been proposed to alleviate the computational burden of fitting complex spatio-temporal hierarchical models. One of such strategies is that applied in INLA. Computationally, INLA is much more efficient as it is based on the use of sparse matrices. Moreover, INLA is based on approximating the marginal posterior distributions (by using Laplace and other numerical approximations and numerical integration schemes) and is usually faster and more accurate than MCMC alternatives. We include these details on the computational costs in the conclusion section of revised paper.*

Minor:

Line 134: Add references for the relation between temperature, wind speed, RH and ozone, PM and NO2.

*References have been included for these parameters.*

**Technical:**

Line 179: 'given in Appendix A'.

*corrected*

Can the authors also add legends to Figure 7 instead of only describing them in the text?

*We will substitute Figure 7 with a more detailed comparison between the results of the statistical model and the observations, extended to all pollutants (not only $PM_{10}$). Anyway, the paper has been revised to include a better description of all figures.*

---

## Author Comment (AC3)

**Reply to referee #1**

1) Section 2.2: It is suggested to provide a table of all the data and their sources in the supplement or other appropriate locations of the paper. Too many URLs are present in this section.

*We thank the referee for the suggestion. We provide a new table in the supplement with information on the data source (see Table A1).*

2) Line 132: Temperature is a very important parameter relevant with photochemistry and lifetime of air pollutants. Why is it not included in the predictor list?

*We thank the referee for pointing out this feature. Temperature was considered as a predictor, but we forgot to list this parameter in the predictor list in Section 2.2. We are sorry for this oversight; in the revised version we specified the use of temperature.*

3) Line 200: Since the sites are unevenly distributed and many non-urban areas are not monitored. I would anticipate significantly reduced site density in Central and South Italy. So, should we also consider more even inclusion/representation of sites in different part of Italy?

*The density of monitoring sites in Central and Southern Italy is reduced compared to that in Northern Italy. In the revised version we included more details on the performance of the model as a function of the characteristics of the measurement stations (urban, urban background, or rural/remote, as classified in the EEA database). Each type of station is represented in each Italian region, and, according to the results shown in the revised paper, we found no differences in performance depending on geographical location. Finally, we underline that stage 2 of our approach is based on the use of different types of information, mainly linked to geographical characteristics and meteorological conditions. This additional information is uniformly distributed throughout the Italian territory and as highlighted in the work, allows an excellent correction of the systematic bias and an adaptation to local conditions. We included a new figure and a new table (see Table 1 and new Figure 3) showing the distribution of the average bias (distinguished by north, center and south stations) for all pollutants.*

4) Section 4.1: I think it is well anticipated that the 11 models will have varying biases and precision. Maybe this section can be moved to the supplement?

*We agree with your suggestion and move Section 4.1 to the appendix.*

5) Table 2: For all the four pollutants and in the "training" and "prediction" rows, the absolute biases are amplified from Step 1 to Step 2. It appears unusual and not found in previous studies. Why and does it matter?

*The first stage is a bas correction step, ie. the b1,…,bm parameters are constructed so as to remove the bias. The second stage introduces new information, i.e. the spatio-temporal covariates. This new information is always beneficial, as shown by the further reduction of the mean squared error (reported in the same table), at the small cost of increasing the mean bias in some cases. In any case, we are talking about rather small deviations. For example, for $PM_{10}$ the bias changes from 0.20 to -0.97 $\mu g/m^3$, which is below 1 $\mu g/m^3$ in both cases. For $PM_{2.5}$ changes from 0.37 to -0.58 $\mu g/m^3$. Similar considerations are also valid for $NO_2$ and $O_3$. These differences are less than one or two orders of magnitude of the typical average concentration values of these pollutants, in line with those reported in other studies. We believe that these errors do not detract from the significance of our statistical treatment, even if at present we cannot exclude a deepening of the nature of this behavior in a future work.*

6) If Figure 2 is only briefly discussed and Table 2 is mainly used in Section 4.2.1, maybe Figure 2 should also be moved to the supplement?

*Figure 2 shows the Taylor diagrams for the validation and prediction dataset, but they are a replica of the same information reported in Table 2 (this is why we did not re-discuss these diagrams in detail). For conciseness, we decided to discard Figure 2 in the revised version.*

7) Section 4.4: NO2 has the strongest spatiotemporal variability due to its short lifetime. I believe case studies using NO2 can provide the most relevant information about model capability. Why is PM10 discussed here? Should similar results for the other pollutants be included in the supplement?

*We expanded the analysis, including:*
1. *a map for the comparison between observations and model values, highlighting the dependence on the type of monitoring station (urban, suburban and rural) and season (see new Figure 3) and geographical region (see new Figure 4).*
2. *a comparison extended to all pollutants (not only PM10, see the new section 4.3)*
3. *Predictions on the regular grid is shown for all pollutants*

8) Figure 8: Please 1) add a map of median of raw predictions and 2) add observed values on the maps. Also, how to assess if the predicted values over unmonitored areas are accurate? Line 346 discussed "extrapolation ability", but quantitative evaluation of such ability is missing. Some "spatial-clustered" cross-validation idea (e.g., doi: s41467-020-18321-y) might be useful.

*We included the box-wisher plots for all pollutants in the revised paper (see new Figures 3 and 4). This analysis has been made to highlight the performance over the dependence on the type of monitoring station (urban, suburban and rural) and season. As explained in Section 3.2 (Validation), we split the whole dataset (about 700 monitoring stations) in two sets. $\approx 90\%$ were used to train the model, the remaining set (the validation dataset) was used for validation purposes (not used during the training phase for the first and second stage). The comparison with the validation data set represents a measure of the expected error in forecasting independent data at "unmonitored" locations.*
*We are sorry but we are unable to resolve the doi number you provided. Missing digits?*

---

## Author Response (AR2)

According to Referee #1, we added an additional sentence (see lines 193-195), to reply to its minor revision recommendation. We thank the reviewer for its suggestion.